# Adaptable Agent Populations
# via a Generative Model of Policies

**Kenneth Derek**
MIT CSAIL
kderek@alum.mit.edu

**Phillip Isola**
MIT CSAIL
phillipi@mit.edu

## Abstract

In the natural world, life has found innumerable ways to survive and often thrive. Between and even within species, each individual is in some manner unique, and this diversity lends adaptability and robustness to life. In this work, we aim to learn a space of diverse and high-reward policies in a given environment. To this end, we introduce a generative model of policies for reinforcement learning, which maps a low-dimensional latent space to an agent policy space. Our method enables learning an entire population of agent policies, without requiring the use of separate policy parameters. Just as real world populations can adapt and evolve via natural selection, our method is able to adapt to changes in our environment solely by selecting for policies in latent space. We test our generative model's capabilities in a variety of environments, including an open-ended grid-world and a two-player soccer environment. Code, visualizations, and additional experiments can be found at https://kennyderek.github.io/adap/.

## 1   Introduction

Quick thought experiment: imagine our world was such that all people acted, thought, and looked *exactly* the same in *every* situation. Would we ever have found the influential dissenters that sparked scientific, political, and cultural revolutions?

In reinforcement learning (RL), it is common to learn a single policy that fits an environment. However, it is often desirable to instead find an entire array of high performing policies. To this end, we propose learning a generative model of policies. At a high level, we aim to show that purposefully learning a diverse policy space for a given environment can be competitive to learning a single policy, while better encompassing a range of skillful behaviors that are adaptable and robust to changes in the task and environment. We name our method of learning a space of *ad*aptable *a*gent *p*olices: ADAP.

Why should we bother with finding more than one policy per environment? We propose two primary reasons. First, RL environments are continually approaching greater levels of open-endedness and complexity. For a given environment, there might be an entire manifold of valid and near-equally high performing strategies. By finding points across this manifold, we avoid 'having all eggs in one basket,' granting robustness and adaptability to environmental changes. In the event of a change, we are able to adapt our generated population to select individuals that can still survive given the ablation, much like natural selection drives evolution in the real world. Secondly, using a generative model of policies as a population of agents makes intuitive sense in multi-agent environments, in which different agents should have the capacity to act like they are unique individuals. However, it is common in many multi-agent reinforcement learning settings to deploy the same policy across all agents, such that they are essentially distributed clones. Doing so may reduce the multi-modality of the agent population, resulting in a single 'average' agent.

Previous work has touched on ideas akin to a generative model of policies. In hierarchical RL, the high-level policy controller can be considered a generator of sub-policies that are 'options' [1, 2, 3]. But these methods are designed to find decomposable skills that aid in the construction of just one

35th Conference on Neural Information Processing Systems (NeurIPS 2021).

downstream controller policy. A core idea of our work is that of quality diversity [4], which aims to optimize a population of agents along the axes of both reward and diversity. Traditional methods often use evolutionary search over a discrete-sized population of separate agents, each with their own policy parameters. This consumes more time and training resources, and limits the number of potential behaviors. Our work integrates the goals of quality diversity into time and memory efficient deep RL by simulating an entire population of agents via a generative model of policies, with diversity bounded only by capacity of the generator.

The rest of the paper is organized as follows. First we introduce our generative model of policies and the diversity objective that guides its learning. Next, we explore the potentials of learning a population of agents by ablating environments and then searching for suitable policies, directly in latent space. We primarily study two environments: Markov Soccer [5] and Farmworld. Farmworld is a new environment we have developed for testing diversity in a multi-agent, open-ended gridworld. At the website linked in the abstract, one can find qualitative results of experiments presented in this paper, as well as additional results on toy environments of CartPole [6] and a standard multi-goal environment.

## 2 Method

Let $\mathcal{Z}$ be a sample space of $n$ dimensional vectors, and $Z$ be a random variable defined uniformly over $\mathcal{Z}$. Then, we learn a mapping, $G : \phi, Z$, from generator weights $\phi$ and latent distribution $Z$ to a space of policies $\Pi$.

The generator $G_\phi$ itself is not a policy. It must be conditioned on a draw $z \sim Z$ in order to define a learned set of behaviors. In this sense, $z$ is a stochastic parameter of $G_\phi$, and is sampled once at the beginning of each agent episode.

In our experiments, $\mathcal{Z}$ is the sample space of all three dimensional vectors with magnitude one (i.e. the surface of the unit sphere). Practically, we use the low dimension of three, so that we can perform a key subject of this paper: rapid optimization, or adaptation, of $G$ by changing $Z$ rather than $\phi$ (fine tuning $\phi$ would be more typical in literature). We require magnitude one so that there is at least one non-zero element for any $z \sim Z$, which we found important for providing signal and stability in the training of $G$. It is possible that with higher dimensions, this stipulation could be relaxed.

Figure 1: Method Diagram. Upon agent initialization, we sample a latent $z_i \sim Z$ which, along with $\phi$, defines an agent policy for the episode. We update $G_\phi$ by optimizing two objectives. The PPO surrogate loss is optimized in an online manner, using the trajectory $\tau$ from the generated policy $\pi_{\phi,z_i}$. Meanwhile, the diversity regularizer loss is optimized over independently sampled policy pairs $\pi_{\phi,z_j}, \pi_{\phi,z_k} \sim Z$.

**Diversity Regularization**  In order to learn a diverse space of unique policies, we introduce a diversity regularization objective. Since policies define a space of actions taken over different states, we propose that in order for two policies to be distinct, they must have different action distributions given the same state. To this end, we define the objective $L_{div}$ (1):

$$L_{div}(\phi) = \mathop{\mathbb{E}}_{s \in S} \left[ \mathop{\mathbb{E}}_{z_i, z_j \sim Z} \exp\left(-D_{KL}(\pi_{\phi,z_i;b}(s) \| \pi_{\phi,z_j;b}(s))\right) \right] \tag{1}$$

in which $D_{KL}$ is the KL-divergence between the two policy action distributions $\pi_{\phi,z_i}$ and $\pi_{\phi,z_j}$, and $b$ is a smoothing constant over the action distributions.

**Optimization of $G$**  In our experiments, we optimize the diversity objective in an online fashion using gradient descent, in conjunction with a PPO [7] clipped-surrogate objective and an entropy regularization objective. Our full optimization problem is

$$\max_{\phi} L_{PPO}(\phi) - \alpha L_{div}(\phi)$$

where $L_{PPO}$ is Equation 9 in [7] and $\alpha$ is a coefficient to scale the diversity regularization objective. See Algorithm 1 in the supplement for additional details.

**Adaptation via Optimization in the Latent Space of** $G$   By learning an entire space of policies $\Pi$, we are able to search our policy space for the highest performing policy, whether dealing with the training environment or an ablated future environment.

In contrast to searching over policy parameters through transfer learning or fine-tuning, we are able to quickly search over the low-dimensional latent space (dimensionality 3 in our experiments). In fact, we can quickly adapt back and forth to various situations: the search procedure often takes *less than 30 seconds*, or 100 episode rollouts, to find any high quality solutions that exist. Over the course of a small number of generations, we evaluate randomly sampled latents, and keep higher performing ones with greater probability. In the event that episodes have a high degree of variablility per run – such as in the Markov Soccer environment – it may be necessary to run several episodes per latent vector and average the returns. Details can be found in Algorithm 2 of the supplement.

**Model Architecture**   Similarly to prior work [3], we have found that richer integrations between the latent vector and the observation can yield a more multi-modal policy space. To induce this richer integration, we introduce a multiplicative model denoted "(x)" for latent integration, and compare the results to a baseline of concatenating "(+)" the latent sample to the observation. We describe this architecture in the supplement.

## 3   Related Work

**Quality Diversity**   The evolutionary computing community has developed various quality diversity (QD) algorithms that aim to find a balance of novel and high-performing individuals within a population. Some methods can even be considered policy generators: NEAT and HyperNEAT [8, 9] use an indirect encoding to construct a network architecture. To encourage diversity, these methods use an idea known as *fitness sharing*: if genotypes are too similar, then they will split reward.

While NEAT and HyperNEAT encourage diversity of *parameters*, other methods encourage diversity of *behavior*. Novelty Search (NS) [10] learns individuals that have high novelty along some user defined behavioral distance metric. For example, in a maze navigation task, the behavioral characteristic could be the final resting location of the individual, and agents are selected based on how far away they end up from an archive of past individuals. Unfortunately, as shown in [11], the choice of this characteristic can critical, and domain dependent. Additionally, NS focuses mainly on finding *novel* solutions, and ignores *fitness*, or reward. NS with Local Competition [12] and MapElites [13] aim to solve this problem by selecting for individuals with high fitness, but only against individuals in the same phenotypic or genotypic region, respectively.

There are several prior and concurrent works that aim to connect ideas of quality diversity with deep reinforcement learning. Like quality diversity algorithms, these methods optimize a fixed-size population or archive of policies to be distinct from each other. [14, 15] aim to find a set of policies that yield diverse trajectories. [15] in particular focuses on the application to multi-agent environments and zero-shot coordination. [16] uses a KL-divergence over policies; but a policy's diversity is optimized over previous SGD updates of itself, thus limiting the potential multi-modality of solutions. [17] optimizes for diversity of the total population via maximizing the determinant of a population distance matrix, but works best only with small populations of size three or five. [18] uses a method reminiscent of DIAYN, but introduces ideas to balance quality with diversity. It is especially similar to ADAP in optimizing the latent space to achieve robustness, but only searches over a fixed-size set of latent vectors and focuses on single-agent environments. Other methods have explored indirectly influencing diversity via differing training hyperparameters as in Population-Based Training [19], or using reward randomization as in [20].

Importantly, both classical QD algorithms [10, 12, 13] and most deep RL methods [14, 15, 16, 17, 19, 20] use sets of distinct agent parameters to learn a diverse population. ADAP makes the connection that we can encode unique policies into a latent space (an idea that also appears in a few recent

works [2, 3, 21, 18]), and frames learning a diverse population as a generative modelling problem. Additionally, in distinction from classical QD methods that use a non-differential genetic algorithm or evolutionary search for optimization, ADAP is able to directly optimize for diversity and policy credit assignment via gradient descent.

**Option Discovery for Hierarchical RL**   The option framework introduced by [1] could be thought of as learning a generator of skills, which are temporal abstractions over actions that can be used by a downstream, higher-level controller. Recent works like DIAYN [2] and others [3, 21] in option discovery learn a fixed set of diverse skills that are discriminable by observed state or trajectory: such as learning to move left, or move right. These skills are generally not meant to be the final agent policy, DIAYN even learns skills without any extrinsic environmental reward. However, these methods are most similar to ADAP in terms of mapping a latent sample to final agent policies.

**Goal-Conditioned Reinforcement Learning**   Yet another way to induce diverse policy behaviors is through using goal-conditioned policies [22, 23, 24] that use a family of task-defined value or Q functions or expert trajectories [25] to incentivize diversity. These methods require structure in how to define diversity, such as defining a value function family over states [24].

**Multi-Agent Roles**   Recent works generate specialized agent policies in a multi-agent setting, building on QMIX [26]. ROMA [27] learns agent roles that are not static through agent trajectories, require optimizing several additional objectives, and are learned jointly with other roles via a joint action-value function. Similarly, MAVEN [28] optimizes the mutual information between joint agent actions and a latent variable. While a single latent sample in ADAP encodes a single agent 'species', a latent sample in these works encode how a group of agents should behave together: thus we cannot employ adaptation based on individual selection.

## 4   Introduction to Farmworld

We test our learning $G$ in a new open-ended grid-world environment called Farmworld, that supports multi-agent interaction and partially observable observations. The idea behind Farmworld is simple: agents move about the map to gather food from various resources, such as chickens and towers that spawn in random locations. In out experiments, agents only optimize their own reward: a single agent gets exactly 0.1 reward for each timestep it is alive. Thus, lifetime is directly proportional to reward. Agents can live longer by attacking other agents, chickens, and towers: for example, a chickens might take two timesteps of sword hits to yield five timesteps worth of health. To avoid cannibalism in our experiments, we set agents to gain zero health from other agents. Of course, these numbers are configurable to achieve different environment dynamics.

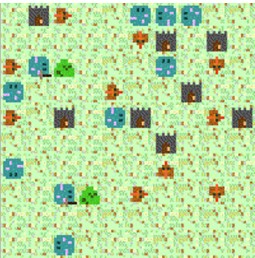

Figure 2: Standard Farmworld Training Environment

Furthermore, Farmworld is a partially-observable environment: agents see only what is in a certain tile radius from their location. In our experiments, the observation is a vector representation of the units and tiles. Additional details of the Farmworld are provided in the supplement.

## 5   Baselines

We use compare the ADAP algorithm to two algorithmic baselines. For each of the baselines, as well as ADAP, we experiment with both concatenation (+) and multiplicative model (x) types, and use consistent observation spaces, action spaces, and latent distributions - so the only difference is the diversity algorithm itself.

The first baseline is Vanilla PPO, which we call the "Vanilla" baseline. The only difference between Vanilla and ADAP is that the former does not use the diversity regularization loss in Equation 1. Vanilla policies still receive samples from latent distribution $Z$ - there is simply no objective term that enforces a diverse policy actions conditional on these samples.

Our second baseline was adapted from DIAYN. DIAYN is formulated as a unsupervised skill generator, rather than a policy generator. However, we believe that it remains one of the technically

closest works, and with slight modifications, we attempt to make a comparison between DIAYN and ADAP. First, we highlight some differences between the methods. ADAP uses a KL-divergence based diversity term rather than learning a skill discriminator network. This enables ADAP's policy diversity to be optimized directly through gradient descent with respect to parameters $\phi$, rather than be optimized through RL as with the skill diversity of DIAYN. Additionally, the ADAP latent distribution is defined over a continuous sample space, in contrast to the categorical sample space of DIAYN. We tried the standard DIAYN algorithm with categorical sample spaces and unsupervised skill discovery, however this performed poorly on all of our Farmworld and Markov Soccer experiments. Thus, to place the algorithms on more equal footing, we modify DIAYN: 1.) add extrinsic environmental reward to DIAYN training (this is briefly mentioned in the DIAYN paper itself) 2.) to use the continuous sample space 3.) train a skill regressor that minimizes predicted latent error, instead of a skill discriminator that outputs latent class probabilities. We describe the new skill regressor in the supplement. We call this method DIAYN*.

*Training and Hyperparameters* We train each method for the same number of timesteps (30 million), and generally keep hyperparameters constant across methods. These are described in the supplement.

*Adaptation Comparisons* When we apply Algorithm 2 to ADAP, *we apply the same algorithm* to each of the baselines. We can do this because ADAP and baselines all share the same input latent distribution $Z$ - the only difference is how well they encode a diverse policy space within $Z$.

## 6    Adaptation to Environmental Ablations via Optimizing $Z$

In nature, differences between species and even within species lend robustness to life as a whole. It becomes less likely that any single perturbation in the environment will break the overall system. In the same manner, differences between policies can lend robustness to the policy space as a whole.

| Ablation | Description |
|---|---|
| Far Corner | 18x18 map size. Food spawns in the bottom right, agents spawn in the top left. |
| Wall Barrier | A 'rift' opens up between agents and their food. Agents must be able to navigate up and around the wall. |
| Speed | Single agent on 2x2 map. Food health yield is set very low, so agents must be able to rapidly and consistently farm adjacent towers. |
| Patience | Single agent on 2x2 map. Food yield is very high, but food respawn speed is low. Agents must 'ration' their food so it lasts until the next respawn. |
| Poison Chickens | Agents spawn on the same map in which they were trained. However, chickens now yield *negative* health. Towers still yield positive health. |
| Training Env. | (Not an ablation: listed for reference) 10x10 map size. Food and agents are uniformly randomly distributed. |

Table 1: Farmworld Ablations

**Experiment**    We aim to test how having a diverse policy space allows us to search in latent space for policies that better fit unexpected environmental ablations. Doing so would demonstrate the robustness of a population of policies, and simultaneously provide information about different types of diversity that are learned by $G$.

To this end, we train $G$ on a normal Farmworld environment as shown in Section 4. We then *ablate* the environment, changing features such as map size and features, location of food sources, and even re-spawn times and food-yield. Lastly, we deploy $G$ into the ablated environment and *without* changing the parameters $\phi$, we optimize the latent distribution for policies that are successful in the new environment, using the search Algorithm 2. Ablations and descriptions are available in Table 1.

**Results**    Rather to our surprise, in each experiment trial, learning $G$ using ADAP created a policy space $\Pi$ containing 'species' that could thrive in nearly every environmental ablation (see Figure 3). The important thing to note is the development of these species was *emergent from the training environment* – a product of optimizing $G$ for both policy diversity and reward maximization.

How is it possible that ADAP produced a policy space capable of adapting to nearly every ablation? The training environment was relatively abundant with resources scattered about a large map. Thus,

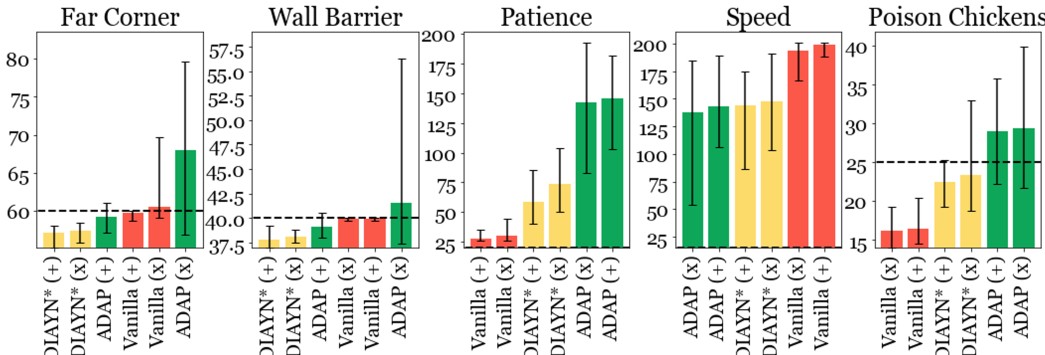

Figure 3: Agent Lifetime *After* Latent Distribution Optimization on Ablations. The bars display the average lifetime over 10 random seeds after applying Algorithm 2 to each of the algorithm and model choices. We additionally plot the **min** and **max** values from each seed pool to indicate that most baselines never come close to solving the ablations, where "solving" is indicated by the average value surpassing the baseline health of the agent (the dashed black line). It is possible to go below initial health, since agents can do damage to each other or eat poisonous food.

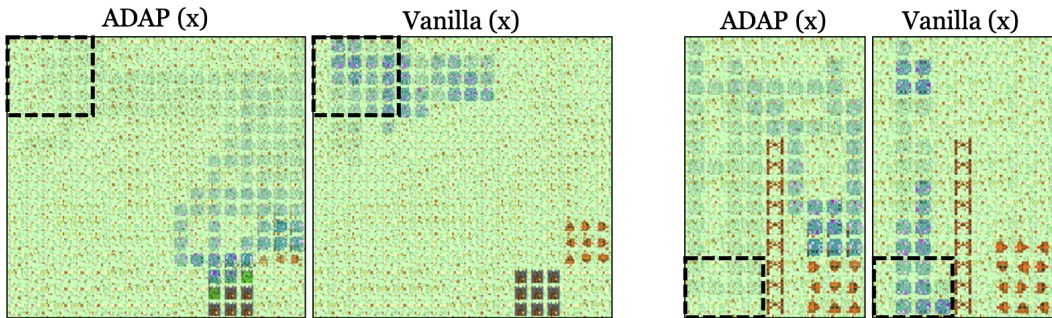

Figure 4: Adaptation on Farmworld Locomotion Ablations. Randomly chosen episode visualizations after latent optimization of `Far Corner` (left) and `Wall Barrier` (right). Chosen ADAP latents yield agent policies that navigate to the food, without otherwise modifying network parameters. Agents spawn in the dotted black outlines, and timesteps are stacked such that earlier ones are more transparent. We show 60 and 40 timesteps for `Far Corner` and `Wall Barrier` respectively.

there were many degrees-of-freedom in the rules of survival, and by optimizing for diversity, we found a policy space that filled these degrees-of-freedom while still yielding high reward. While these ablations reflect some of the possible axes of diversity, there are certainly more. For example, an agent's direction of 'preference' does not have to be the bottom-right, as in the `Far Corner` ablation. Indeed, as a sanity check, we tested placing food locations in various other spots on an enlarged map, and found that for every cardinal location, there was a species of agent in $G$ that could exploit that new food location. What came as a surprise was that agents also used their health indicator to diversify: since agents diversify conditional on state, species developed in which agents would prefer to go upwards when their health is high, but downwards when their health is low. This particular

| | Vanilla (x) Initial | Vanilla (x) Fine Tuning - 2000 Episodes | ADAP (x) Latent Optimization - 200 Episodes |
|---|---|---|---|
| Wall Barrier | 40 | 40 | 47.5 |
| Far Corner | 60 | 60 | 74 |
| Patience | 28.5 | 55.2 | 158 |

Table 2: Latent Distribution Optimization Speed and Performance versus Fine Tuning. Notice that for some of the difficult exploration problems, the Vanilla model ever even finds the ablated food location, even with 10 times the amount of training data. On the other hand, ADAP's diverse policy space enables commited exploration via selecting species that have tendencies to move towards different parts of the map.

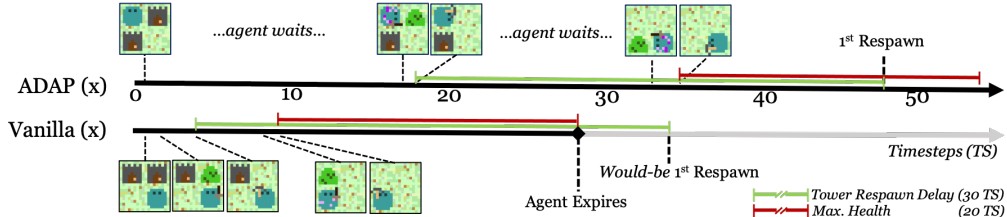

Figure 5: Adaptation on Farmworld `Patience` Ablation. We show episode rollouts arranged along a timeline, using agent policies after latent optimization. Notice that the Vanilla agent does not ration out the two towers. Meanwhile, the selected ADAP agent waits until its health is very low (visualized by purple and pink speckles) before getting food and is able to survive past the next tower respawn.

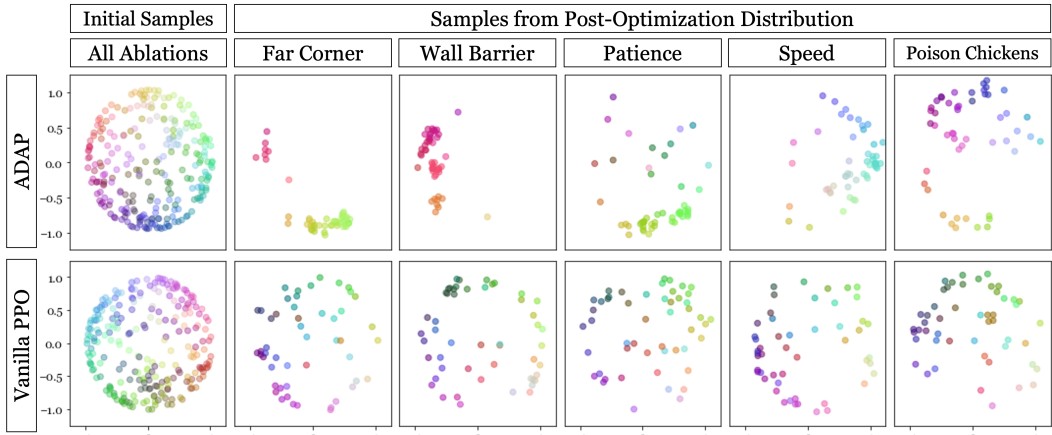

Figure 6: Pre and Post Optimization Latent Distribution Samples per Ablation. For each ablation, we run Algorithm 2 on a randomly chosen ADAP and Vanilla generator, $G_{\phi,Z}$. We refer to the post-optimization latent distribution as $Z'$. For visualization, we then sample $z \sim Z'$ for 50 times for each ablation, and use PCA to reduce the dimensions of each method's data. The RGB color of each sample is directly proportional to the initial 3-dimensional sample. Notice how the for the Vanilla generators, the post-optimization distributions match the initial distribution, indicating that no regions of the latent space were found to be well-suited for any particular ablation. However, with ADAP generators, we see that the resulting distribution $Z'$ is vastly different in many cases to the initial distribution $Z$. This analysis also reveals certain overlap between two ablations. Notice how the $Z'$ for the Wall Barrier ablation has samples that in some places overlap with Far Corner samples. Intuitively, in Far Corner, agents that move right-and-then-down are successful, while in Wall Barrier, agents that move up-then-right-then-down are successful.

agent species was the one that managed to thrive in the `Wall Barrier` ablation. Similarly, in the `Patience` ablation, ADAP learned a certain species of agent that waited until its health was low before farming a tower.

The `Poison Chickens` ablation was the one hold-out in which latent optimization on ADAP could not find a profoundly successful species. It is possible that, there would have been too large of a trade-off between diversity and potential reward in the training environment in order to learn a policy that ignored half of its potential food sources. We come back to this ablation in the next experiment.

Finally, we should note that ADAP beat the Vanilla baseline in all ablations aside from `Speed`. We hypothesize this ablation is the most in-distribution to the training environment. Since the Vanilla baseline optimized for solely for expected rewards, it makes no diversity tradeoffs and performs well in in-distribution environments. As visible from the plots, DIAYN* also did not learn to speciate in a manner that was successful on the majority of ablations.

## 7   Measurement of Agent Individuality and Diversity in a Population

A good generative model of policies should be able to represent a multi-modal space of behaviors. That is: different agent policies should be able to act with individuality. Our generative model uses a

|            | Lifetime       | $I(T;Z)$ | $H(T)$ | $H(T\|Z)$ |
|------------|----------------|----------|--------|-----------|
| ADAP (x)   | **23.2** ± 0.5 | **0.89** | 1.0    | 0.10      |
| Vanilla (+)| 22.6 ± 0.3     | 0.30     | 0.99   | 0.69      |
| Vanilla (x)| 22.5 ± 0.4     | 0.32     | 0.99   | 0.67      |
| ADAP (+)   | 21.9 ± 0.6     | 0.76     | 0.92   | 0.15      |
| DIAYN* (x) | 21.8 ± 1.3     | 0.58     | 0.95   | 0.37      |
| DIAYN* (+) | 21.6 ± 0.4     | 0.49     | 0.93   | 0.44      |

Table 3: Various Specialization and Diversity Metrics. We provide the average result across 10 random seeds, and ± indicates standard deviation. Standard deviation for $I(T;Z)$, $H(T)$, $H(T|Z)$ was less than or equal to 0.1 for all entries, and has been omitted to reduce verbosity. $H(T)$ is computed with log base 2, and is maximal at 1.

shared parameter set across all agents, and naively using shared parameters could result in learning just one 'average' agent – which is precisely what we wish to avoid.

**Niche Specialization Experiment**   To test the abilities of our policy generator, we set up the Farmworld environment with a hidden rule specific to this experiment: when an agent spawns, it is able harness resources from either towers or chickens. However, once it gets health from one unit type, it becomes 'locked-into' that unit type, and cannot gain health from the other unit type. Information about an agent's 'locked-into' state is not provided as part of the agent observation, and since agents have no memory, they would have to look to their latent $z$ to determine their niche. Since there are equal numbers of chickens and towers on our map, a reasonable generative model algorithm should be able to map half the latent space to each of these two specializations, or niches.

**Results**   So we can see how well the *entire* latent space falls maps to a niche, we report rewards and other metrics in Table 3 *without* running latent space optimization on ADAP or baselines. In summary, ADAP consistently learned a more multi-modal policy space than any of the other baselines. Our results also indicate that using a multiplicative model can yield a higher degree of policy space multi-modality, and therefore greater success in this environment.

We can see in Table 3 that ADAP (x) is able to attain the highest average agent lifetime. This, however, is not necessarily the most interesting point. ADAP, learns a policy generator with the highest mutual information $I(T|Z)$ between an agent "niche" $T$ and the latent distribution $Z$. Intuitively, this means that ADAP was able to learn a population of agents that were composed of two clear species – on one hand: agents that focus on chickens, and on the other: agents that focus on towers. Formally, let $T$ be a discrete random variable where $p_T(t)$ is the probability that an agent attacks target $t$, for $t \in \{chicken, tower\}$. Then $I(T;Z)$ is high when individual agents are specialized in a niche, and we see diverse niches across our population. This is because $I(T;Z) = H(T) - H(T|Z)$ and is maximized by both increasing $H(T)$ and decreasing $H(T|Z)$. $H(T)$ measures the *diversity* of niches across all agents in the population, and $H(T|Z)$ measures how rigidly an agent falls into a single niche (i.e. *specialization*).

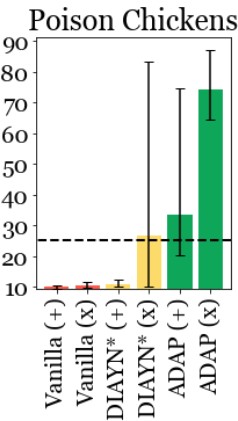

Figure 7: Poison Chickens Revisited. We report min, max, and average lifetime after running Algorithm 2.

As an example, suppose agents were highly specialized but not diverse, e.g., all agents were chicken-only attackers. Then $H(T) = H(T|Z) = I(T;Z) = 0$. On the other hand, suppose that all $z \sim Z$ yield an agent policy that attacks chickens and towers with equal probability. Then in this case $H(T) = H(T|Z) = 1$ and $I(T;Z) = 0$. Intuitively, this means half of the time agents are wasting timesteps to attack a target that they are unable to even damage! Qualitatively, we have seen that the latter case occurs with the Vanilla and (most seeds of) DIAYN* baselines: notice that their $H(T|Z)$ is significantly higher than that of ADAP.

For fun, we performed latent distribution optimization on generators trained using the Niche Specialization environment to fit the Poison Chickens environment. One would expect algorithms with high $H(T|Z)$ to fare well, since Algorithm 2 can find optimized $Z^*$ such that $p_T(chicken|z \sim Z^*) = 0$. Sure enough, we see this result in Figure 7: ADAP (x) is most suc-

cessful at consistently producing a generative model that can produce policies that not only avoid chickens, but also successfully attack only towers.

# 8 Adaptation and Self-Play in a Zero-Sum Two-Player Environment

**Environment**   This experiment uses Markov Soccer, introduced in [5]. Two agents, A and B, play on a gridworld and must 'carry' a ball into the opposing goal to score. Agents walk in cardinal directions or `stand` in place. Possession is randomly initialized, and switches if one an agent bumps into the other. Actions of A and B occur on the same timestep, execution order is randomized, and each timestep ends in a draw with some $\epsilon$ probability.

Markov Soccer is an interesting environment, because the best policy for one agent depends on the policy of the other agent. As described in [5], there exists a worse-case-optimal probabilistic policy for Markov Soccer, which maximizes the minimum possible score against any adversary. This strategy tends to be conservative, preferring to act towards a draw where a different policy could have obtained a higher score. On the other hand, non-worse-case-optimal strategies may be less conservative and may achieve very high scores against some opponents, but very low scores against others. Analogous to real soccer, different players have varying abilities and play styles, and a given player $p_1$ may be optimal against $p_2$, but not against $p_3$.

If any single policy has its drawbacks, can we instead learn an entire space of diverse policies $\Pi := \{\pi_1, \pi_2, ..., \pi_{\inf}\}$, where for any opponent, we can select a policy $\pi_i \in \Pi$ that achieves the maximum score against that opponent? Ideally, this space includes the worse-case-optimal policy, as well as other more aggressive policies. Then, just as a coach might swap out a soccer player, we can mix and match our champion as suited.

**Experiment**   Can we learn a population of individuals that is holistically strong against all types of opponents? We evaluate adaptability to various adversaries using two methods. First, we test baselines and our method against a set of hand-coded soccer bots. These bots are designed to represent a wide gamut of strategies, some of which are more exploitable than others. Secondly, we evaluate each $G$ by playing ADAP (x), ADAP (+), Vanilla (x), and Vanilla (+) in a round-robin tournament against each other. All scores is determined by wins minus losses over 1000 simulated games.

*Against Hard-Coded Bots:* Each bot always starts on the left side, and the learned policy starts on the right side (although the environment is coded such that observations are side-invariant). Bot types fall into three categories: offense (bots start with possession), defense (policy starts with possession), and mixed (random starting possession). See Table 4 for more details.

| Bot | Bot Type | Bot Policy |
|-----|----------|------------|
| `Straight` | Offense | Always moves `right`, towards the goal. |
| `Oscillate 0` | Defense | Oscillates in column 0, blocking both squares adjacent the goal. |
| `Oscillate 1` | Defense | Oscillates in column 1, leaving a gap in front of the goal. |
| `Stand` | Defense | `stands` in one square adjacent the goal, leaving the other open. |
| `Rule-Based` | Mixed | Follows hand-coded heuristics. |
| `Random` | Mixed | Follows a random policy. |

Table 4: Markov Soccer Bot Adversaries

*Round-Robin Against Each Other:* We also pit each generative model in a round robin tournament against the other models. The manner in which we do this is described in the supplement.

**Training and Baselines**   We use self-play to train both ADAP and baselines. We use the same Vanilla baseline as described in Section 5, and we omit the DIAYN* baseline for brevity. Note that *at no point* in the training process did any of our algorithms train against any bots, or against each other.

**Results**   As in the Farmworld adaptability experiment, we see from Figure 8 that ADAP is able to learn a $G$ during the *train* phase that emergently contains members that are successful against a variety of unexpected adversaries - including naive bots and other policies.

Compared to Vanilla, the ADAP policy space generalizes better against all adversaries. Going back to the soccer team example, we were able to select individuals from the ADAP population that were well suited for a specific strategies. For example, against the `Oscillate 1` adversary, ADAP latent

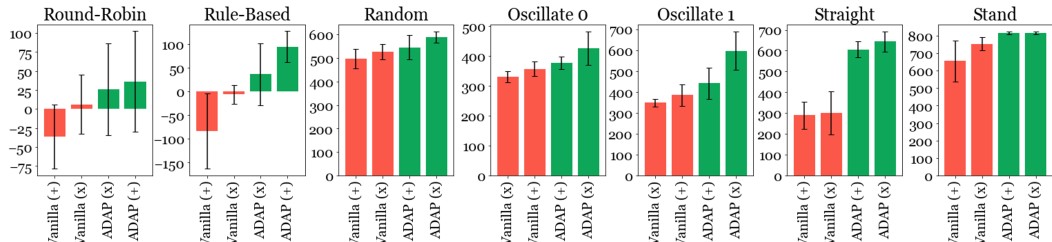

Figure 8: Results of round-robin and bot games. Score is over 1000 runs, some games prematurely end in a draw. We report mean and standard deviation over five random seeds.

optimization found a member of the population that *side-stepped* the oscillating adversary simply by moving to the top row, and then down to the goal. Additionally, against the `Straight` adversary, successful ADAP individuals stole possession by deterministically standing in-front of the opponent to block, and then moving around and into the goal. On the other hand in both of these situations, Vanilla could not find individuals that exploited the naive deterministic nature of their opponents.

Using ADAP did not just allow us to optimize against naive opponents. ADAP learned the best $G$ in the round-robin tournament, and was the only method that was able to consistently beat our rule-based bot. It is possible that by using ADAP during the self-play training, individuals encountered a wide variety of strategies that bettered overall performance.

## 9 Limitations

**Bad Apples**   When using ADAP, not every member of the policy space is going to be an optimal policy. In fact, some generated policies might be bad apples: policies that were incentivized by the diversity regularizer to take actions that were not rewarding. Naturally, some individuals might be better or worse than others. These individuals can be removed by optimizing the latent distribution. However, the bad apples may come with a plus side. Even though they do not perform well in the current environment, they might happen to perform well in a future ablated environment!

**Continuous-Action Space Environments**   The results presented so far focus entirely on environments with discrete categorical action spaces, in which we have observed that our diversity regularizer in Equation 1 empirically performs well. However, not all environments in RL use discrete action spaces - continuous action spaces are widely used in RL control tasks. While we believe that our regularizer can work in these environments, we have not rigorously tested in these environments.

## 10 Conclusion

We have presented a framework to learn a generative model of policies. Rather than learning just one policy, we aim to find as many high-performing and individually distinct policies as possible, all compressed within the parameters of our generator.

Learning a space of policies pays off in an open-ended environment such as Farmworld, in which there may be more than one path to success. We show in Section 6 that we can adapt to ablations by quickly choosing 'species' from our learned policy space that are successful in the new environment.

We also learn a policy space in a competitive, two-player, zero-sum game in Section 8. Here, no single deterministic policy is optimal against all adversaries. Instead, we show how to train a family of policies that can be naturally adaptable to a wide array of both challenging and naive adversaries.

Overall, we hope to show how it can be beneficial in RL to optimize not just for reward, but also for diversity of behavior. As environments continue to increase in complexity and open-endedness – filled with branching paths to success – it makes sense to learn not just one, but many, solutions.

## 11 Acknowledgements

This research was supported in part by IBM through the MIT-IBM Watson AI Lab.

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
