# 1 Supplement

## 1.1 Model Architectures

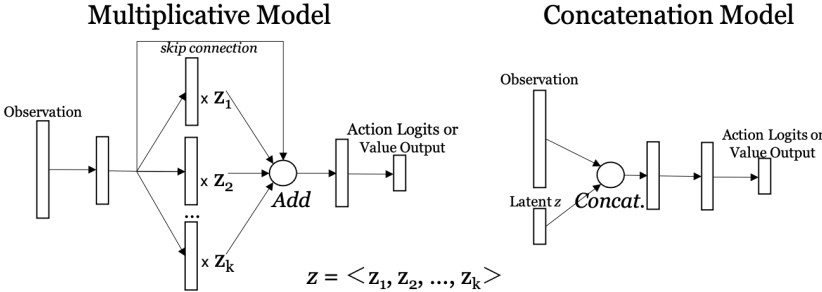

Figure 1: Model Architectures for Latent Integration

Using a latent vector of dimension $k$, our multiplicative model is able to learn $k$ interpretations of the observation, which are each modulated by a dimension of the latent vector. A skip connection allows the model to learn policies faster than without. As a baseline, we use a concatenation model, in which the latent vector $z$ is concatenated with the environment observation at each timestep. In both cases, by setting corresponding model weights to zero, a learned policy could completely ignore the latent vector to yield a standard RL policy architecture.

Note that the multiplicative model architecture comes with increased computational cost, in which for a hidden dimension of size $d$ and latent dimension $k$, the number of parameters of the hidden layers are bounded by $\Theta((k+1)d^2)$, whereas in the concatenation model, they are bounded by $\Theta(d^2)$. In practice, since $k$ and $d$ are small ($k = 3$ and $d \in \{16, 32, 64\}$) in our experiments, the increase in computational cost is not significant.

## 1.2 Algorithm Pseudocode

---
**Algorithm 1** ADAP with PPO

---
1: $m$ the number of sampled latents in diversity estimation
2: $n$ the number of sampled states in diversity estimation
3: $k$ latent vector size
4: $\alpha$ diversity regularization coefficient
5: **for** for iteration = 1, 2, ... **do**
6:     Let $B$ be an empty batch of (s, a, r) tuples
7:     **for** actor $a$ = 1, 2, ..., $N$ **do**
8:         Sample latent $z$ from latent distribution
9:         $B \leftarrow$ Run policy $\pi(\cdot|\theta_{old}; z)$ in environment for $T$ steps
10:         Compute advantage estimates $\hat{A}_1, ..., \hat{A}_T$
11:     **end for**
12:     Sample $M \in \mathbb{R}^{m \times k}$ from the latent distribution         ▷ latent matrix
13:     Sample a batch $S$ of $n$ states from $B$
14:     $L_{div} \leftarrow 0$
15:     **for** $i = 1, 2, ..., m-1$ **do**
16:         **for** $j = i+1, i+2, ..., m$ **do**
17:             $L_{div} \leftarrow L_{div} + \frac{1}{n}\sum_{s \in S} D_{KL}(\pi(s|\theta_{old}, M^{(i)}), \pi(s|\theta_{old}, M^{(j)}))$
18:         **end for**
19:     **end for**
20:     $L_{div} \leftarrow \frac{2}{m(m-1)}L_{div}$         ▷ Scale by number of policy-distance pairs
21:     Maximize $L_{PPO} - \alpha L_{div}$ w.r.t. $\theta$ via SGD.
22:     $\theta_{old} \leftarrow \theta$
23: **end for**

---

**Algorithm 2** Latent Distribution Optimization
```
 1: Input: g the number of optimization generations
 2: Input: E an environment
 3: Input: G a policy generator
 4: Input: Z a latent distribution with dimension k
 5: Initialize: best ← descending sorted array
 6: for i = 1, 2, ..., g do
 7:     explor ∼ Unif([0, 1])
 8:     r ∼ Unif([0, 1])
```
9:     **if** ($\texttt{explor} \leq 0.5$ **and** $i \leq \frac{3}{4}g$) **or** $\texttt{len}(\texttt{best}) \leq 10$ **then**
10:         **if** $r \leq 0.5$ **or** $\texttt{len}(\texttt{best}) \leq 10$ **then**
11:             $z \sim Z$           ▷ Random Sampling
12:         **else**
13:             $z \leftarrow \texttt{sample}(\texttt{best[0:10]})$
14:             $z \leftarrow z + \text{project}_Z(\text{Unif}[\text{-}0.1, 0.1]^k)$       ▷ Mutation
15:         **end if**
16:     **else**
17:         **if** $r \leq 0.5$ **then**
18:             $z \leftarrow \texttt{sample}(\texttt{best[0:10]})$       ▷ Replication
19:         **else**
20:             $z \leftarrow \texttt{pop}(\texttt{best[0:10]})$       ▷ Pruning
21:         **end if**
22:     **end if**
23:     $score \leftarrow$ Reward from running $\pi_{G,z}$ on $E$
24:     $\texttt{best.push}(z)$ with key $score$
25: **end for**
26: **Return** $\texttt{best[0]}$

## 1.3 Description of Farmworld

Farmworld is an open-ended gridworld environment designed with two goals in mind: high customizability and support for diverse solutions. The environment is written entirely in Python, and is easily hackable. Maps can be hand-crafted, or randomly generated. At a high level, agents must traverse the environment to find resource units, and harvest health from these resources. Agents can interact directly by attacking each other, or indirectly by competing for shared and limited resources.

Units have configurable levels of health, damage, food yield, and respawn, which can be utilized to encourage learning a variety of different policies. For example, by placing low food yield on resource units (chickens and towers) and high food yield on agent units, one may incentivize direct multi-agent competition (agents must attack each other to get health). Conversely, setting a high resource unit health can encourage multi-agent co-operation - agents will have to work in parallel to mine a chicken or tower before their health runs out.

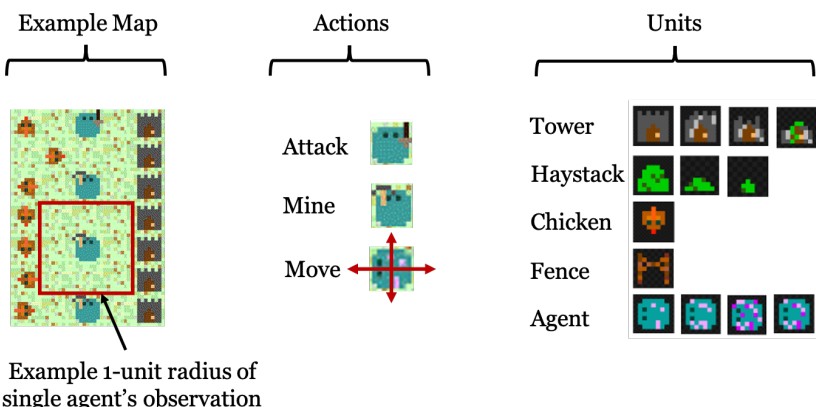

Figure 2: Illustration of Farmworld and Units

**Observation Space**   Can be either RGB images, or a flattened array of unit-encoding vectors. If RGB images are used, agents 'see' exactly what we see: units visibly lose health by damage patterns that appear over time, and unit orientations can be discerned by the unit 'eyes' (see the Example Map in Figure 2). If unit-encoding vectors are used, then all units have encoded `health`, `orientation` (by default, 0 - 3 to represent each possible cardinal direction), and unit `type` (e.g. 0 for ground, 1 for agents, 2 for chickens, 3 for towers, 4 for `fences`). Encodings are scaled to be within [0, 1].

Agents have partially-observable observations: they do not see the entire map. By default, they can see units in a L1 radius of 2 unit squares.

**Action Space**   The action space is 6-dimensional categorical, respresenting `up`, `down`, `right`, `left`, `attack`, `mine`. Actions can be added or removed as necessary.

**Unit Pecularities**   towers and chickens each have a corresponding non-negative `respawn_time`. chickens disappear after they get hit `max_health` times by an agent `attack`. Unlike towers, chickens are able to move 1 square in any direction on each timestep, with `chicken_move_probability`.

tower units are more tricky:  they turn into a `haystack` in the same location after `max_chicken_health` of `attack`. Haystacks must be `mined` with a pickaxe, and only after `max_tower_health` will they yield food resources.

`Fence` units are simple: they cannot be destroyed or moved. Additionally, no units can pass through them.

**Reward**   Agents get an individual reward of 0.1 for each timestep that they are alive. If an agent's health reaches 0, it is removed from the map and other agents can carry on as normal. The entire episode ends when `max_episode_timesteps` is reached, or when no more agents are alive on the map.

## 1.4 Round Robin Tournament in Markov Soccer

Let $S(\pi_a, \pi_b)$ be the score of player $\pi_a$ in one round of the game against player $\pi_b$. Then to find the score of a generator $G_1$ versus $G_2$, we attempt to find the best policy of $G_1$ with respect to the best policy of $G_2$ against $G_1$ on average. Formally, let

$$z_2^* = argmin_{z_2} E_{z_1 \sim Z}[S(\pi_{\phi_1, z_1}, \pi_{\phi_2, z_2})]$$
$$z_1^* = argmax_{z_1} S(\pi_{\phi_1, z_1}, \pi_{\phi_2, z_2^*})$$

Then, the score of $G_1$ versus $G_2$ is $S(\pi_{\phi_1, z_1^*}, \pi_{\phi_2, z_2^*})$ over 1000 games. The final score of $G_1$ is the average of ($G_1$ versus $G_2$) and ($G_2$ versus $G_1$).

## 1.5 Baselines

DIAYN [1] originally attempts to maximize the mutual information between the state and a discrete categorical latent vector by optimizing an intrinsic reward generated from discriminator error. We wanted to make a comparison of DIAYN to ADAP in which both methods used continuous latents to find a potentially unbounded number of niches. To this end, we augmented DIAYN and called this DIAYN*. In DIAYN*, we train the discriminator to regress the latent, rather than predict the latent category. We add this intrinsic reward to the extrinsic environmental reward, giving us the new reward function $r'$:

$$r_t' = err_t + r_t$$

where

$$err_t = -\alpha(q_\phi(s_t) - z)^2 - \text{mean}(err_{batch})$$

$z$ is the latent vector, $\text{mean}(err_{batch})$ is the mean discriminator error across the update batch, and $\alpha$ is the scaling of the intrinsic reward (generally set at 0.05). We subtract by the batch mean so that on average, the expected agent reward equals only what is provided by the extrinsic environment. Otherwise, original DIAYN and DIAYN* stuggled with balancing dense extrinsic environmental rewards from the experiment with the intrinsic discriminator reward. In our niche specialization experiment, we also experimented with the canonical DIAYN. In this implementation, we use categorical contexts and we add extrinsic reward directly to intrinsic discriminator reward. As mentioned in the paper, this method did not perform well in our Farmworld Niche Specialization experiment.

Finally, we treat DIAYN and DIAYN* like a generative model of policies (since we are not trying to learn options). To do so, we keep $z$ fixed throughout an agent episode. Included in the website are toy experiments that benchmark our implementations of DIAYN*.

### 1.6 Smoothing Parameter $b$

In continuous domains with action distribution $\mathcal{N}(\mu, \sigma)$, we observed that the KL-divergence in Equation 1 may encourage very low $\sigma$ values early in ADAP training. To solve this, we used standard deviation $\sigma' = \sigma + b$, where $b$ is a small constant (ex: 0.05). We similarly use the smoothing parameter $b$ in the discrete action spaces, but have not tested whether or not it is necessary in these situations.

### 1.7 Training Hyperparameters

Unless otherwise mentioned, we used optimized our policies using a clipped PPO surrogate objective with learning rate 3e-4. Advantages were computed using Generalized Advantage Estimation, with a $\gamma$ discount factor of 0.99, a $\lambda$ smoothing parameter of 1, and a gradient clip of 0.5. We use the RLLib [2] framework for training, using their default PPO configuration. For all experiments, we use concatenation and multiplicative model architectures as seen in the main paper. Importantly, we always use *separate value and policy networks*. Attempts to combine these networks generally resulted in non-diverse policy spaces, which we believe is a result of the importance of the value function in recognizing the differing expected rewards conditional on each latent from the latent space.

For multi-agent environments, batch sizes are always in *agent* steps, rather than in *environment* steps. Thus, if there are 40 agents in an environment, then 1 environment step is 40 agent steps.

To optimize our diversity regularization objective, we use parameters $m = 10, b = 30, k = 3$, as detailed in 1. However, preliminary investigation into the effect of these hyperparameters indicates that it is possible to get away with even smaller samples of latent vectors and states, while still effectively optimizing for a diverse policy manifold.

Unless otherwise specified we use a diversity regularizer coefficient on our novel objective of coefficient of 0.1 in CartPole, 0.2 in Farmworld, 0.2 in Markov Soccer, and 0.5 in MultiGoal. When using DIAYN and DIAYN*, we found that a small intrinsic reward coefficient of 0.05 was best. Anything beyond that, and DIAYN and DIAYN* had issues optimizing for actual extrinsic reward.

For all methods, we generally use an 0.05 entropy coefficient, except in Markov Soccer in which we also run Vanilla PPO with 0.1 entropy coefficient and were able to achieve slightly stronger performance. In our Markov Soccer experiment, we report the average of these two Vanilla results.

| | |
|---|---|
| Batch size | 4000 |
| Minibatch size | 400 |
| SGD iterations per batch | 10 |
| Training epochs | 200 |
| Hidden dimension | 16 |
| Value Activations | ReLU |
| Policy Activations | Tanh |

Table 1: CartPole

| | |
|---|---|
| Batch size | 4000 |
| Minibatch size | 400 |
| SGD iterations per batch | 10 |
| Training epochs | 500 |
| Hidden dimension | 32 |
| Value Activations | ReLU |
| Policy Activations | Tanh |

Table 2: Multi-Agent MultiGoal

| | |
|---|---|
| Batch size | 8000 |
| Minibatch size | 8000 |
| SGD iterations per batch | 10 |
| Training epochs | 10 thousand |
| Hidden dimension | 64 |
| Value Activations | ReLU |
| Policy Activations | Tanh |

Table 3: Niche Specialization and Farmworld Ablation Experiment

| | |
|---|---|
| Batch size | 8000 |
| Minibatch size | 8000 |
| SGD iterations per batch | 10 |
| Training epochs | 10 thousand |
| Hidden dimension | 64 |
| Value Activations | Tanh |
| Policy Activations | Tanh |
| GAE lambda | 0.95 |
| GAE gamma | 0.9 |

Table 4: Markov Soccer