# OpenReview forum: "Adaptable Agent Populations via a Generative Model of Policies"
_NeurIPS.cc/2021/Conference — NeurIPS 2021 Poster_

### Official Review · Reviewer_Q4DY · 2021-07-05

**Rating:** 6
**Confidence:** 3

**Summary:**

This paper proposes to train a generative model for entire populations of maximally diverse agents, from which one specific individual policy can quickly be selected at deployment time through a fast search process.

Policies are represented as networks augmented with a low-dimensional latent variable z, randomly sampled at agent initialization. Thus each trained network is actually a generative model, from which an infinity of policies can be generated by sampling over z.

Crucially, the training procedure encourages the fixed weights of the network to not only obtain good performance for any given random z, but also to produce maximally different policies for any two different z. Thus the method is a quality diversity (QD) method (though the authors seem ambiguous on the subject).

This method is compared with a previously published method (DIAYN), and with itself but without the diversity objective, and is found superior to both on various gridworld tasks, both in terms of expected performance and diversity of generated agents.

**Limitations And Societal Impact:**

Limitations and impact seem adequately addressed.

**Main Review:**

I find the method interesting and somewhat novel (see below). Building an infinite, continuous space of diverse but efficient agents, through which one can quickly search (and thus adapt) at deployment time, as opposed to the discrete populations typically maintained in QD algorithms, seems like a significant advantage.

One possible concern is insufficient relation with previous work. Bluntly: Am I correct that  this method is basically DIAYN with continuous 3D z and a different, simpler diversity objective? If not, why?

Note that this would not be a deal-breaker, since DIAYN is already used as a baseline and found less efficient for the tasks selected here. But somehow it's not mentioned in the "related work" section! It seems important to explain how this method relates to DIAYN and how it differs from it (e.g. in objectives, motivation, etc.), in addition to the results.

The paper is confusingly written and some passages were unclear:

- Most importantly, in p. 5, the description of baselines (especially lines 171-173) is extremely obscure and it took me a while to understand (?) what was meant. I suggest replacing the start of l. 173 with the following: "the Vanilla baseline also includes a latent z; the only difference between 'Vanilla PPO' and ADAP is that the former does not use the diversity regularization (Eq. 2)". If this is incorrect, an equivalent explanation of what exactly Vanilla PPO means should be provided. Also: "DIAYN* uses a continuous distribution" - over what ?? (I suppose it should be over "skills", which are very similar to the latent z).

- The farmworld environment should be described a little bit more in the main text, e.g. what's the difference between towers and chickens, what does "attacking" another agent mean, etc. Just pointing to the supplements is not enough.

- In p.8, "G1 vs G1" ? What is pi_G1,z1 ? Also, this evaluation method seems obviously asymmetric: G1 is forced to produce a generalist, while G2 can choose an agent that specifically exploits G1, presumably lending an advantage to G2. How is this asymmetry handled?

- Whenever the z is optimized, it should be mentioned at least briefly how, *in the main text*.

Minor points:

- In "related work" sections: The proposed method is very much a quality diversity method, by definition.Probability distributions over actions *are* a behavioral characterization, and KL divergence between them is a novelty metric. It may be different from existing QD methods, but it's still one.

- p.2, line 60: does (1) mean Equation 1? It should be spelt out.

- Typos: p. 4, l. 163: missing "to". p. 5, l. 172: missing ")". p. 6, l. 227: what's "the 2"?

**Time Spent Reviewing:**

4

---

> ### Author Response · Authors · 2021-08-10
> **Response to Reviewer Q4DY**
>
> Thank you for your feedback! We have worked on revising the related works section to indicate how this method is also a QD algorithm. We will try to address specific points you’ve made below:
>
> > Unclear baselines:
>
> Apologies about the confusion, and thank you for working through it! Your understanding of Vanilla PPO is correct. Regarding DIAYN* -- DIAYN* samples its context from a continuous uniform distribution (the same Z distribution that ADAP uses). How it “chooses” to encode the context distribution into skills is up to the training process.
>
> We’ve been working on making the baseline explanations clearer.
>
> > Farmworld explanation
>
> Thanks for the suggestion, we will add explanation of the key details of the environment in the main paper (around line 158), and will also release the environment as opensource code with full documentation on a GitHub repository.
>
> > G1 vs. G1
>
> “G1 vs. G1” should instead be “G1 vs. G2”
>
> $\pi_{G1,z_1}$ should instead be $\pi_{\phi_1,z_1}$ (to maintain consistency with the notation in Figure 1). In the revision we will just replace any $G_n$ with $\pi_{\phi_n,z_n}$.
>
> You’re right about the potential asymmetry - our method is such that if we have generators 1, 2, and 3, the score of 1 is avg(1vs.2, 2vs.1, 1vs.3, 3vs.1).
>
> > Mentioning Z latent distribution optimization in main text
>
> To adapt to ablations, we optimize the latent distribution in the same manner (using Algorithm 2) -- regardless of whether we are using ADAP, DIAYN, or Vanilla baseline. This is why we describe the procedure once in 87-90. Please let us know you think this needs further clarification in the main paper. The description is also currently high level overview since it is not the primary focus of the paper, but we will try to incorporate more specific details.
>
> > Am I correct that this method is basically DIAYN + mention of DIAYN in related works
>
> First, we believe that one of our primary contributions is showing how a generative model of policies is applicable to fast adaptation and learning multi-agent populations. Our experiments in Markov Soccer also indicate success regarding training using self-play. We believe these ideas introduce novelty in comparison to DIAYN.
>
> Indeed there are strong technical similarities to DIAYN, and we mention it in the related works section on line 107. But there are some noteworthy technical differences:
>
> - DIAYN primarily focuses on learning skills without regard to an external reward function.
> - A continuous z allows for a policy space that can be as large or small as necessary, theoretically finding as many ways as possible to be diverse, and potentially enabling interpolation of learned policies.
> - Our simpler diversity objective has several benefits:
>      - No need to have a skill discriminator network, which solves the problem of what is used as input to the discriminator network (DIAYN uses state, but there are many possible choices: VALOR explores using trajectory + bidirectional RNN discriminator)
>      - We can optimize the diversity objective directly via gradient descent, whereas DIAYN uses RL by providing an intrinsic discriminator-based reward
> -DIAYN uses a concatenation model type, while we explore concatenation and multiplicative
>
> We will clarify both the similarities and the differences in the revision.
>
> > does (1) mean Equation 1?
>
> Yes, thank you for pointing this out.

---

> > ### Comment · Reviewer_Q4DY · 2021-08-22
> > **Thank you**
> >
> > Thank you for your response. These important clarifications should be included in the paper. Conditional on this, and on satisfaction of other reviewers' requests, I believe the paper is acceptable.

---

### Official Review · Reviewer_Pf7s · 2021-07-12

**Rating:** 6
**Confidence:** 3

**Summary:**

The authors proposed a generative model of policies, which maps a low-dimensional latent space to an agent policy space to learn a space of diverse and high-reward policies on any given environment (without requiring the use of separate policy parameters). The proposed method is able to adapt to changes in our environment solely by selecting policies in latent space. The experiments evaluated our generative model’s capabilities in a variety of environments, including an open-ended grid-world and a two-player soccer environment.

The strength of this paper is as follows: 1) The proposed method integrates the goals of quality diversity into deep RL by simulating an entire population of agents via a generative model of policies. 2) The authors evaluated this method using three different experiments and showed that this method was able to learn a more multi-modal and effective policy space than any of the other baselines.


**Limitations And Societal Impact:**

Limitations and societal impact were mentioned.

**Main Review:**

However, there were some unclear points. First, although the authors describe the novelty of this work in the Related work section fragmentally, there was little description of the technical novelty. In my understanding, it seems to be the diversity regularization and multiplicative model, but for those, there was no description of the related work in Section 2 (since the ideas are simple, it would be necessary to quote other work). In addition, although the experimental results were better, the theoretical contribution of this work should be clarified. Lastly, as described in abstracts, time and memory efficiency were not theoretically and numerically analyzed.

Minor comments:

Eq. (1): S and s were not defined. If s is a state, actions may not be also defined.

111 “requiring discriminability via state or trajectory impose diversity constraints along a and speciﬁc task-dependent axis. “ Does the author have theoretical grounds for that? Or just understanding from the experimental results?

172 no parenthesis

176 I did not understand why the authors augment the intrinsic reward with extrinsic environmental reward.


**Time Spent Reviewing:**

5 hours

---

> ### Author Response · Authors · 2021-08-10
> **Response to Reviewer Pf7s**
>
> Thank you for your feedback, we will try to address specific comments below:
>
> > Technical novelty
>
> As you mention, we introduce a new diversity regularization, and experiment with different stochastic neural network architectures (although SNNs themselves are not new to the field).
>
> A large part of the novelty is connecting Quality Diversity with deep RL, under the framework of a generative model of policies. As opposed to learning a fixed sized set of policies, we offer a way to learn a potentially unbounded sized space of policies.
>
> Finally, a part of the contribution is to show the applicability to
> - fast adaptation
> - using adaptability as a way to measure policy space diversity
> - connections to multi-agent learning (niche specialization experiment)
> - self-play (markov soccer experiments)
>
> > Quoting relevant related work
>
> There are several referenced works in the Related Works section, largely in the option discovery (line 105) and quality diversity (line 124) areas.
>
> > Theoretical contribution
>
> Our focus of this paper is to bring attention to the applicability of using a generative model of policies to different areas in deep RL. While theoretical contributions are valuable and certainly may follow in future works, we believe the initial best way to do this is through empirical success.
>
> > Time + memory efficiency in abstract
>
> You may be referring to line 30 in the introduction? Here, we contrast our method to using $n$ different parameter sets -- one for each agent. Analyzing memory efficiency in terms of parameter size is possible:
>
> - Multiplicative model: the number of parameters of the hidden layers are bounded by $\Theta((k + 1)d^2)$ where $k$ is the latent size and $d$ is the hidden dimension
> - Concatenation model: number of parameters bounded by $\Theta(d^2 )$.
>      - In practice, $k$ and $d$ are small ($k$ = 3 and $d$ $\in$ {16, 32, 64})
>
> Thus, in large multi-agent simulations that use $n$ separate models of hidden dim d, the parameters are bounded by $\Theta(n * d^2)$, whereas our method will use a fixed $\Theta((k + 1)d^2)$ or $\Theta(d^2)$. Since $n$ is the only term that is non-constant, our method uses asymptotically fewer parameters.
>
> Practically, training is faster on our method, since you can batch across all agents during training.
>
> > Diversity along task-dependent axis
>
> This sentence could use clarifying, thank you for pointing it out. A better term may be “discriminability-dependent axis.” A simple example is considering the difference between discriminating latent contexts via state vs. via trajectory: in the former, visitation order does not matter, but this does not hold true for the latter.
>
> > Augmenting intrinsic reward with extrinsic reward
>
> Most experiments in DIAYN focus on unsupervised skill discovery to learn skills that are later applicable for a hierarchical controller. However, we want to find quality skills that are also diverse. Here, quality is defined by the extrinsic reward function.
>
> Take a 2D gridworld goal reaching example. Without using an extrinsic reward function, DIAYN will find skills that cover the full region of the grid, whether or not they reach the goal. To encourage DIAYN to find diverse ways to reach the goal, we augment the intrinsic reward with extrinsic reward.

---

> > ### Comment · Reviewer_Pf7s · 2021-08-31
> > **Thank you for the response**
> >
> > I understood some of my questions, but I am concerned about clarity of the theoretical contribution and specific method selection in Section 2. Totally, I evaluated their technical novelty so I increased the score.
> >
> > > There are several referenced works in the Related Works section, largely in the option discovery (line 105) and quality diversity (line 124) areas.
> > I already confirmed the related work in Section 3. My intention about the comment is that in Section 2, the authors mentioned little about why they selected such a method, and just described what the authors did. I am unfamiliar with some pieces of related work, so I cannot understand some of the specific technical motivations in Section 2.

---

### Official Review · Reviewer_xPri · 2021-07-16

**Rating:** 6
**Confidence:** 4

**Summary:**

This paper presents ADAP, a method for learning a generative model of diverse policies by conditioning a standard PPO policy on an additional latent z input, such that KL divergence between the z-conditioned policies for pairs of different z's is maximized. The experiments show that ADAP is able to learn a diverse set of policies

**Limitations And Societal Impact:**

- The authors acknowledge that their existing experiments relate to simplistic discrete-action settings.
- I am curious whether this method can scale to much more complex environments, and whether low-dimensional latents are still sufficient to capture the useful diversity in behavior for the environment.

**Main Review:**

ADAP is a simple method that seems to produce a diverse of set of policies. The experiments are on conducted on two interesting environments that are not commonly studied in deep RL. While the method itself seems quite interesting and the experimental results are promising, I believe the paper could be strengthened considerably by improving the presentation of results, as well as filling in a few key missing empirical considerations that would clarify the utility of the method in practice:

- In the niche specialization experiment, it seems that ADAP leads to higher mean agent specialization, but there is no indication of whether the specializations themselves are more or less diverse than those produced by DIAYN. Some quantitative measure of this would add confidence to the idea that ADAP is effectively optimizing for diversity.
- It would be informative to report how many episodes are required for adapting the pretrained policy by searching over the latent vectors z. A useful comparison here would then be to give Vanilla PPO the same budget of environment interactions for fine-tuning its policy to the new ablated environment. Similarly for adapting to Markov Soccer opponents.
- Further, since searching over latents can be seen as a form of weight optimization, another fair comparison would be to train vanilla PPO with a dummy latent z concatenated to its input, then performing the same search procedure in Algorithm 2 to adapt the vanilla PPO policy via search over of its latent input vector.
- It would raise confidence in the results to report results for additional seeds (5-10 seeds).
- It is hard to parse the relative diversity of the learned latent policies between DIAYN and ADAP from the current results. Some way of globally summarizing the differences in diversity among the agent populations trained via ADAP and DIAYN would be informative. For example, perhaps training agents in a 2D goal-reaching grid world environment, and showing the average state occupancy of rollouts sampled from each latent population, after various time steps.
- It would be valuable to better understand how modulating the weight of the ADAP diversity bonus and the dimensionality of the latent encoding z impacts the transferability of the optimal latent policies to transfer environments.

Additional comments:
- It is unclear to me what the dotted lines in Figure 3 represent. Are these standard deviations around the mean?
- Was Algorithm 2 also applied to finding the optimal skills from DIAYN for adapting to Farmworld variants in the experiment baselines?
- A highly related work should be included in related works: Lupu et al, 2021, [http://proceedings.mlr.press/v139/lupu21a.html](http://proceedings.mlr.press/v139/lupu21a.html).

### Post-rebuttal update
Based on the authors' clarifications and reporting results for additional experiment seeds, I have decided to upgrade my score to a 6.

**Time Spent Reviewing:**

5

---

> ### Author Response · Authors · 2021-08-10
> **Response to Reviewer xPri**
>
> Thank you for your helpful comments, and time spent reviewing this work. We will also work to incorporate suggested related works. Specific points addressed below:
>
> > Niche specialization: indication of the specialization diversity + quantitative measure
>
> You’re right -- the measurement in the paper only captures how specialized individual agents are. But then, a population of chicken-only farmers would actually score high on this per-agent specialization measurement, which is not what we want! Qualitatively, we can see that these methods do not exhibit this behavior, but we will revise our metric to quantitatively assure readers, too:
>
> Let $T$ be a discrete random variable of what unit type an agent will attack with values {chicken, tower}. We can compute the mutual information between $T$ and an agent latent $z$: $I(T; z) = H(T) - H(T | z)$
>
> Our former per-agent specialization measurement treats $H(T)$ as 1, which is actually close to the real observed values. We report here the updated measurement, which we shall call “population diversity”
>
> |             | Population Diversity | Reward |
> |-------------|----------------------|--------|
> | ADAP (x)    | 0.78                 | 18.6   |
> | ADAP (+)    | 0.60                 | 18.3   |
> | DIAYN (x)   | 0.49                 | 17.9   |
> | DIAYN (+)   | 0.34                 | 17.7   |
> | Vanilla (x) | 0.32                 | 17.6   |
> | Vanilla (+) | 0.31                 | 17.8   |
>
> > how many episodes are required for adapting the pretrained policy + fine-tuning Vanilla PPO
>
> Certainly! Preliminary results indicate ADAP is faster at adaptation than fine-tuning PPO, and in some cases is able to find solutions when PPO could not.
>
> For our adaptation procedure (algorithm 2), we give a budget of 200 episodes in Farmworld, although it can usually recover a good latent far sooner (line 92). This equates to just one or two gradient updates in PPO fine-tuning which is hardly effective at all, so instead we fine-tune for 100k environment steps →
>
> PPO average lifetime after fine tuning for 100k environment steps -- which equates to different numbers of episodes, depending on the ablation:
>
> |               | PPO lifetime before fine-tuning | PPO lifetime after fine-tuning 100k steps | Fine-tuning equivalent # of episodes | ADAP lifetime after 200 episodes latent optimization |
> |---------------|------------------------|----------------------------------|--------------------------|---------------------------------------------|
> | Wall Barrier  | 40                     | 40                               | 2500                     | 47.5                                        |
> | Far Corner    | 60                     | 60                               | 1650                     | 74                                          |
> | Patience/Slow | 28.5 +/- 1.6           | 55.2 +/ 12                       | 3000                     | 158                                         |
>
>
> As you can see, fine-tuning on Wall Barrier and Far Corner never even finds the food resources. Fine-tuning on Patient/Slow ablation eventually teaches the agent to wait between attacking food, but at a far slower rate than selecting a “patient” policy from ADAP via latent optimization.
>
> > train vanilla PPO with a dummy latent z concatenated to its input + performing the same search procedure
>
> Agreed - this is what we do in the Vanilla PPO baseline experiments (line 173), and the policy can then choose to use or ignore its concatenated latent. We are working to improve our explanation of these baselines so it is clearer for future readers!
>
> > It would raise confidence in the results to report results for additional seeds (5-10 seeds).
>
> Also agreed, for the revision, we plan to increase the number of seeds for the main experiments.
>
> > hard to parse the relative diversity + training agents in a 2D goal-reaching grid...showing the average state occupancy
>
> We explore a variant of the 2D grid-world goal environment briefly in the supplement (at the bottom of the provided index.html), and find that ADAP and DIAYN generally both cover a large area of the grid. Ultimately, state coverage is just one potential metric of diversity, and does not cover state visitation order, choice of actions that do not result in state changes, etc.
>
> It certainly would be nice to have some way of saying that DIAYN has 70% diversity, where ADAP has 90% diversity (or some other numbers). But in open-ended environments, it is difficult to get these exact measurements. Our best attempt is to see how each algorithm handles the various ablations we throw at it: more diverse algorithms should be able to handle more ablations.
>
> > Impact of diversity coefficient + dimension latent encoding $z$ on the transferability of the optimal latent policies to transfer environments
>
> We conducted ablation studies of the diversity coefficient in Farmworld and Markov Soccer. In Farmworld there is little impact, until reaching a point where it is too high such that it no longer focuses on reward optimization. In Markov Soccer, increasing the coefficient can increase performance in a relatively linear manner on the Oscillating and Straight bots, while performing relatively constant on the Rule-Based, Random, and Stand bots.
>
> We loosely investigated using dimensions of 3, 4, and 6, and found no significant differences, but that’s not to say that further ablation studies could reveal more insightful results.
>
> > Figure 3 dotted lines
>
> Standard deviation about the mean, which we will mention in the revision.
>
> > Was Algorithm 2 also applied to finding the optimal skills from DIAYN for adapting to Farmworld variants in the experiment baselines?
>
> Yes - so that we could make a fair comparison, we used the same Algorithm 2 configuration to attempt to find optimal latents for all ADAP, DIAYN, and Vanilla PPO methods.
> This was in all experiments except for niche specialization, where we did not use Algo. 2 for any method.

---

> > ### Comment · Reviewer_xPri · 2021-08-17
> > **Thanks for the responses**
> >
> > > We report here the updated measurement, which we shall call “population diversity”
> >
> > Thanks for looking into additional analysis aimed to quantify diversity. The mutual information between the attack choice and latent, $I(T;z)$​​​​​​​​​​ indicates the reduction in uncertainty (information gain) about the agent's unit preference given the value of z—so it only captures the specificity in attack choice for a particular agent indexed by z. It seems equivalent to a specialization metric. Therefore I'm not convinced this new metric measures population diversity. Why not look at H(T)  over the episodic distribution of T's obtained for z's sampled at the start of each episode for each method?
> >
> > > Also agreed, for the revision, we plan to increase the number of seeds for the main experiments.
> >
> > Would it be possible to report at least 5 seeds in this discussion (ideally 10) to instill more confidence in these results? Three seeds seems quite low. It seems like the method is fast to run, especially given that you were already able to investigate latent z's with dimensions 4 and 6, mentioned in the rebuttal.
> >
> > > It certainly would be nice to have some way of saying that DIAYN has 70% diversity, where ADAP has 90% diversity (or some other numbers). But in open-ended environments, it is difficult to get these exact measurements. Our best attempt is to see how each algorithm handles the various ablations we throw at it: more diverse algorithms should be able to handle more ablations.
> >
> > I respectfully disagree with this statement. Firstly, none of the environments studied are actually open-ended. Further, for simple environments like goal-reaching grid worlds and CartPole, there are clear ways to quantify the diversity of behaviors. Your left and right reward experiments are examples of this, though since the different settings here are limited, we can only conclude that ADAP performs similarly to DIAYN.
> >
> > Thanks for the clarification on the dotted lines in Figure 3 and when Algorithm 2 was used.
> >
> > I like the method, as it seems simple and competitive with DIAYN. It also seems quickly adaptable to new settings. Some of the analysis seem a bit rough and several parts of the paper could benefit from clearer exposition. The fact that all results only report 3 seeds also concerns me. **More seeds replicating the results would instill more confidence in me and push me over the fence to accept this paper with the expected improvements promised by the authors in their rebuttal.** For now I plan to keep my rating to 5.

---

> > > ### Comment · Reviewer_crDt · 2021-08-20
> > > **Mutual information**
> > >
> > > I'd like to note that $I(T;z)$ can actually become negative if z is a concrete value. Are the authors aware of this and is this conforming with their interpretation?

---

> > > > ### Author Response · Authors · 2021-08-23
> > > > **Negative I(T; z)**
> > > >
> > > > Our understanding is that in fact mutual information is always non-negative [1]. To clarify, in case there was misunderstanding, by z we mean the random variable representing agent latents, rather than a concrete draw of this random variable. That is, I(T;z) is mutual information, rather than pointwise mutual information (which indeed can be negative). Can the reviewer please clarify their concern if this does not address it?
> > > >
> > > > [1] "Elements of Information Theory", Cover & Thomas, 1991

---

> > > > > ### Comment · Reviewer_crDt · 2021-08-26
> > > > > **z**
> > > > >
> > > > > Using a low caps $z$ indicates a concrete value. You should make *very* clear that this is actually a random variable. Notation-wise, this is quite unfortunate, I would strongly recommend writing capital $Z$ instead if at all possible.
> > > > >
> > > > > This can throw a reader off quite badly, as $I(T;z)$ reads like a point-wise/local quantity.

---

> > > > > > ### Author Response · Authors · 2021-08-26
> > > > > > **Low caps $z$**
> > > > > >
> > > > > > Agreed, when we include the new mutual information quantities in the paper, we will be sure to revise the notation. $\mathcal{Z}$ for the sample space, $Z$ for the random variable, and $z$ for a sample of the random variable.

---

> > > ### Author Response · Authors · 2021-08-23
> > > **Additional seeds and I(T;z)**
> > >
> > > Thanks for your additional comments. We address each point below, and will update with additional results asap.
> > >
> > > > Seeds
> > >
> > > We are running experiments with 5 or more seeds now and will report results as soon as we have them (we expect it will take a few more days).
> > >
> > > > Population diversity metric
> > >
> > > We argue that I(T;z) is a measure of both agent specialization and population diversity. This is because I(T;z) = H(T) - H(T|z), and is maximized by both increasing H(T) and decreasing H(T|z). H(T) measures the diversity of behaviors across all agents in the population and H(T|z) measures the diversity of behaviors of a single agent — low H(T|z) is what we mean by “specialization” of a policy.
> > >
> > > As an example, suppose agents were highly specialized but not diverse, e.g., all agents were chicken-only attackers. Then H(T) would equal H(T|z) -- knowing z yields no information gain about the behavior. Now suppose each agent is either chicken-only or tower-only but an equal number play each role. In this scenario, H(T|z) would be the same value as before (p(T|z) is a one-hot distribution in both cases) but H(T) is now maximal (p(T) is a uniform distribution). Hence I(T;z) is higher in the latter scenario and all that changed was that the population level diversity increased.
> > >
> > > One can also construct the scenario where H(T) is maximal but agents are not at all specialized, i.e. p(T|z) is uniform. Then H(T) = H(T|z) and I(T;z) = 0. Here we see that losing specialization also decreases I(T;z).
> > >
> > > In sum, I(T;z) measures both diversity and specialization and is maximized when both are high. We therefore maintain that I(T;z) is a good metric but our naming of this metric could be improved. In the revision we will include the table below, where we show H(T), as a measure of observing diverse behaviors, H(T|z), as a measure of specialized behaviors, and I(T;z) as a single quantitative summary of both specialization and diversity.
> > >
> > > |            | I(T;z)                 | H(T)                  | H(T\|z)                |
> > > |------------|------------------------|-----------------------|------------------------|
> > > | ADAP (x)   |   0.7869516859075480   |   0.9473035989876250  |   0.1603519130800770   |
> > > | ADAP (+)   |   0.6047703632465650   |   0.9460772684298130  |   0.34130690518324800  |
> > > | DIAYN* (x) |   0.49911232347050400  |   0.9389550705433290  |   0.43984274707282500  |
> > > | DIAYN* (+) |   0.3400399616634490   |   0.9510767303035150  |   0.611036768640066    |
> > > | Vanila (x) |   0.3209150012343460   |   0.967539620324802   |   0.646624619090456    |
> > > | Vanila (+) |   0.31642290800441500  |   0.958975827869018   |   0.6425529198646030   |
> > >
> > > > 2D goal reaching diversity
> > >
> > > We agree that CartPole and goal-reaching grid world should not really be called open-ended and will avoid referring to them as such. The goal of these environments is primarily to serve as a baseline, to show that ADAP and DIAYN work to some extent in creating a population of different solutions. We note that both ADAP and DIAYN* perform similarly in these simple worlds.

---

> > > > ### Comment · Reviewer_xPri · 2021-08-24
> > > > **Response to clarifications**
> > > >
> > > > Thank you for agreeing to run more seeds.
> > > >
> > > > >We argue that I(T;z) is a measure of both agent specialization and population diversity. This is because I(T;z) = H(T) - H(T|z), and is maximized by both increasing H(T) and decreasing H(T|z). H(T) measures the diversity of behaviors across all agents in the population and H(T|z) measures the diversity of behaviors of a single agent — low H(T|z) is what we mean by “specialization” of a policy.
> > > >
> > > > After reviewing your clarification, I agree mutual information seems like a satisfactory measure for diversity and specialization—especially with your presentation which also shows the individual contributions of the entropy and conditional entropy terms.
> > > >
> > > > My remaining question here is what is the outcome space for T? My understanding is that it is either chicken or tower as niches, but then the maximum entropy H(T) should be the uniform distribution over two outcomes, which has a maximum entropy of $\log 2 = 0.693...$, which is lower than your reported H(T) values, which are close to 1.

---

> > > > > ### Comment · Reviewer_crDt · 2021-08-26
> > > > > **log**
> > > > >
> > > > > Are we perhaps talking about binary log? In which case the values would be 1 for uniform.

---

> > > > > ### Author Response · Authors · 2021-08-26
> > > > > **Log base 2**
> > > > >
> > > > > We're glad with the additional clarifications are helpful, and will include those in the revision. Regarding your question, we used log base 2 to compute the entropy, which has a max value of 1 in this case.

---

> > > > > > ### Comment · Reviewer_xPri · 2021-08-26
> > > > > > **Thanks for the response**
> > > > > >
> > > > > > Of course! That makes a lot of sense.

---

> > > > ### Comment · Reviewer_xPri · 2021-08-26
> > > > **More seeds**
> > > >
> > > > After the post-rebuttal discussion, I am happy to increase my score contingent on the results being updated with 5-10 seeds. Looking forward to seeing the updated results in this discussion.

---

> > > > > ### Comment · Area_Chair_sV1y · 2021-08-31
> > > > > **Re: More seeds**
> > > > >
> > > > > Dear Authors,
> > > > >
> > > > > Please let us know if you manage to update the results suggested by reviewer xPri.
> > > > >
> > > > > Reviewer xPri will decide whether they will change their assessment of this work, contingent on you doing so.
> > > > >
> > > > > Best,
> > > > >
> > > > > Area Chair

---

> > > > > > ### Author Response · Authors · 2021-08-31
> > > > > > **Please see our earlier comment to xPri with additional seed results**
> > > > > >
> > > > > > Dear AC,
> > > > > >
> > > > > > In case you didn't see, we replied earlier today in a separate thread to xPri with results of additional seeds: https://openreview.net/forum?id=73FeFxePGc&noteId=jwzeZmpuiZ9
> > > > > >
> > > > > > We are continuing to work on updating the remaining experiments with additional seeds, and will include the full results on 8 seeds in the final paper.
> > > > > >
> > > > > > Best,
> > > > > > Authors

---

> ### Author Response · Authors · 2021-08-31
> **More Seeds**
>
> Hello,
>
> We've run 8 seeds, per method, per model on the Farmworld Niche Specialization experiment. These are entirely separate to the ones we've ran in the paper. The results support the findings in the paper, with $I(T;z)$ very similar to the paper.
>
> We're still working on increasing seeds for the other experiments. Since the Niche Specialization experiment had the most overlap in 'standard-deviation intervals' between the ADAP and the baselines, we thought it was best to do it first.
>
> A few things of note:
> - We run this experiment for fewer steps (6 million vs. 30 million). This was largely due to machine memory and time constraints.
> - Methods generally converge on $I(T;z)$ after ~3 million steps, (hence the numbers are very similar to those in the paper).
> - Reward increases even after $I(T;z)$ converges. In the charts below, reward is similar across methods, but this is because of the fewer training steps. If we run things for longer, we consistently see ADAP continue to improve in reward, whereas "unspecialized" methods remain constant.
> - DIAYN* exhibits similar behaviors as in the paper, where some seeds converge to a specialized population, and others do not. Overall, it has lower mean and min $I(T;z)$ across seeds.
>
> Mean Results
>
> |            | Reward    | H(T)     | H(T\|z)  | I(T;z)   |
> |------------|-----------|----------|----------|----------|
> | ADAP (x)   | 17.090500	 | 0.998972 | 0.175044 | 0.823928 |
> | ADAP (+)   | 16.321750 | 0.938952 | 0.179404 | 0.759548 |
> | DIAYN* (x) | 17.130375 | 0.979037 | 0.335206 | 0.643831 |
> | DIAYN* (+) | 16.459556 | 0.931935 | 0.483371 | 0.448565 |
> | Vanila (x) | 16.790375 | 0.997937 | 0.647089 | 0.350848 |
> | Vanila (+) | 17.109333 | 0.992963 | 0.684358 | 0.308605 |
>
> Standard Deviation of Results
>
> |            | Reward   | H(T)     | H(T\|z)  | I(T;z)   |
> |------------|----------|----------|----------|----------|
> | ADAP (x)   | 0.252078 | 0.001535 | 0.035123 | 0.036294 |
> | ADAP (+)   | 0.281518 | 0.031990 | 0.051633 | 0.049787 |
> | DIAYN* (x) | 0.960680 | 0.021259 | 0.152614 | 0.173490 |
> | DIAYN* (+) | 0.179311 | 0.013413 | 0.025341 | 0.020926 |
> | Vanila (x) | 0.202019 | 0.002090 | 0.015703 | 0.016085 |
> | Vanila (+) | 0.093093 | 0.005340 | 0.017838 | 0.023022 |
>
> Min of Results
>
> |            | Reward | H(T)     | H(T\|z)  | I(T;z)   |
> |------------|--------|----------|----------|----------|
> | ADAP (x)   | 16.898 | 0.995939 | 0.126345 | 0.764548 |
> | ADAP (+)   | 15.733 | 0.876497 | 0.108236 | 0.674660 |
> | DIAYN* (x) | 16.162 | 0.949976 | 0.183306 | 0.465394 |
> | DIAYN* (+) | 16.181 | 0.914685 | 0.443226 | 0.417704 |
> | Vanila (x) | 16.543 | 0.993074 | 0.619766 | 0.336680 |
> | Vanila (+) | 17.007 | 0.987930 | 0.668693 | 0.284157 |
>
> Max of Results
>
> |            | Reward | H(T)     | H(T\|z)  | I(T;z)   |
> |------------|--------|----------|----------|----------|
> | ADAP (x)   | 17.578 | 0.999990 | 0.231391 | 0.873522 |
> | ADAP (+)   | 16.630 | 0.980490 | 0.255890 | 0.833343 |
> | DIAYN* (x) | 18.268 | 0.999666 | 0.484582 | 0.814993 |
> | DIAYN* (+) | 16.705 | 0.958571 | 0.528308 | 0.479557 |
> | Vanila (x) | 17.140 | 0.999204 | 0.662472 | 0.379302 |
> | Vanila (+) | 17.189 | 0.998565 | 0.703773 | 0.329871 |

---

> > ### Comment · Reviewer_xPri · 2021-08-31
> > **Score change**
> >
> > I thank the authors for running more seeds. I upgraded my score to 6, and I will further upgrade to 7, assuming the remaining experiment results also hold up under the additional seeds.

---

### Official Review · Reviewer_crDt · 2021-07-18

**Rating:** 7
**Confidence:** 5

**Summary:**

The paper constructs a latent space diversity-based family of policies
able to adapt to different environments under different
circumstances. The idea is to have a relatively low-dimensional set of
adaptation parameters which essentially are all that needs to be
selected to choose between pre-trained policies.

**Limitations And Societal Impact:**

Reasonable listing of limitations.

**Main Review:**

I do like the premise of the paper, a simple, but promising idea. It
addresses the issue that training is slow, but sometimes one still
needs to adapt quickly to a new situation. The idea of using a
low-dimensional latent space to select from otherwise pre-trained
policies is nice. The reviewer was wondering why, since they already
cited work by Stanley, they didn't refer to MAP elites which is in
spirit even closer (albeit on the EA side of the formalisms) to the
present paper.  More generally, it might be good to relate to
multiobjective optimization, and also to multitask MDPs to make clear
how the present algorithm relates to existing solutions. It is clear
that the diversification is not based on the concrete task set, but on
diversification in latent space, still the latent space is a function
of the training in given sample spaces.

- Eq. (1):
  - what distribution is the inner expectation computed over?
    The notation as a set is really not clear, and z_i != z_j is weird,
    because in continuous distributions that probability were 0. Please
    make clear what you are doing here.
  - you talk about a smoothing constant b, but we have no idea what
    your regularizer looks like. How does the smoothing look like?
- line 227: "using the search algorithm described in the 2." -
  incomplete sentence
- line 235: "emergent from the train environment" -> "training"
- Figure 5: colors need explaining
- line 256: -> training
- line 288: not entirely clear what you do here: are you measuring the
  distance between the two sets of possible policies? Or what else?
  Perhaps write a formula, to avoid ambiguities?
- line 293: the second G_1 should be a G_2


**Time Spent Reviewing:**

2

---

> ### Author Response · Authors · 2021-08-10
> **Response to Reviewer crDt**
>
> Thank you for the helpful comments and insight into prior works. We do actually mention MAP-Elites (line 124) but agree it may be worth further elaborating on connections to ADAP, and including works related to multi-objective optimization.
>
> Regarding specific comments:
> > Eq. (1):
> > Inner expectation distribution:
>
> Z is the Uniform distribution over a $k$ dimensional unit-sphere. We will work on clarifying this notation for the revision!
>
> > smoothing constant b
>
> We do this when sampling actions during training. For categorical action spaces of size $d$, let the output of the final softmax layer be $\{p_0, ..., p_d\}$ that defines the categorical action distribution $X_1$. Then before computing the KL divergence between two categorical action distributions given the same state, we recompute each $p_{X_1}(x=i) = \frac{p_i + b}{1 + db}$ for a given action distribution $X_1$, and similarly for $X_2$. Here, $b$ is a small constant 0.05. We will include this in the appendix of the revision.
>
> > Figure 5: colors need explaining
>
> In the bottom row, the pink lines and hearts indicate where the agent’s health would run out, if it were to do nothing else. This is to illustrate that the Vanilla agent will expire before the next tower respawn point (since it was ‘impatient’), but the ADAP agent can survive past the next tower respawn. The supplement has actual videos of these episodes, and we have since made a new diagram that hopefully is clearer!
>
> > G_1 should be a G_2
>
> Yes, thank you.
>
> > distance between the two sets of possible policies?
>
> Not quite - we are trying to compete two policy generators against each other, in order to see which one is better. To do this, we find latents from each generator and compete them against each other. Specifically, we use the following equations to score the policy generators:
>
> $S(\pi_a, \pi_b)$ is the score of player $\pi_a$ in one round of the game.
> $$z_2^* = argmin_{z_2} E_{z_1 \sim p_z} [S(\pi_{\phi_1,z_1}, \pi_{\phi_2,z_2})]$$
> $$z_1^* = argmax_{z_1} S(\pi_{\phi_1,z_1}, \pi_{\phi_2,z_2^*})$$
> $$G_n(z_n) = \pi_{\phi_n,z_n}$$
>
> and the final score of G1 vs. G2 is $S(\pi_{\phi_1,z_1^*}, \pi_{\phi_2,z_2^*})$ We report the average results of G1 vs. G2 and G2 vs. G1 for symmetry.
> We will include these details in the revision.

---

> > ### Comment · Reviewer_crDt · 2021-08-26
> > **MAP elites**
> >
> > Ok, you do indeed cite MAP Elites literature, maybe say it explicitly (if there is space)?

---

> > > ### Author Response · Authors · 2021-08-26
> > > **QD**
> > >
> > > Thanks for the suggestion, we will revise the related works section to make the connection to map elites and other QD methods clearer.

---

> > > > ### Comment · Reviewer_crDt · 2021-08-31
> > > > **Thanks**
> > > >
> > > > Thank you very much for adding the clarifications. This concludes my questions.

---

### Decision · Program_Chairs · 2021-09-27

**Decision:**

Accept (Poster)

**Comment:**

Inspired by the diversity produced by evolution, this work presents a method of learning a latent space of policies that is conditioned by an RL algorithm like PPO. They show that their model can learn a diverse set of policies, and since the latent space is not too high dimensional, it is easy to adapt to changes in the environment. They demonstrate the model in a grid-world “farm” environment and a two-player soccer environment (I also downloaded the zip file from the supplementary materials, and I found the web page useful to understand some results from the animations.).

The strengths of the paper (best summarized by reviewer Pf7s):

1. The proposed method integrates the goals of quality diversity into deep RL by simulating an entire population of agents via a generative model of policies.

2. The authors evaluated this method using three different experiments and showed that this method was able to learn a more multi-modal and effective policy space than any of the other baselines.

There were issues with the writing and clarity, raised by reviewer Q4DY. The authors have responded to the questions, and have pledged to clarify issues with the paper discussed in the thread.

Reviewer xPri, who wrote a critical, but fair and balanced review, also highlighted a list of important points in their review related to not only improving presentation of the results, but also filling in key missing experiments, and improving statistical confidence in the results. The authors performed several additional studies, and after a long and detailed dialog between reviewer and author (this is where Open Review really shines), an understanding is established and the reviewer is satisfied, improving their score to 6 (even to 7 if additional seeds can be run on remaining experiments).

In summary, this paper presents a simple and promising idea for creating diverse adaptable agent populations in RL; a work that would be of interest to the NeurIPS community. I believe the review process has helped strengthen the paper to a state that we are happy to publish the work at NeurIPS. For these reasons I'm recommending acceptance of the work as a poster.